# Chinese Named Entity Recognition Based on Knowledge Based Question Answering System

**Didi Yin [1], Siyuan Cheng [1], Boxu Pan [2], Yuanyuan Qiao [2], Wei Zhao [1] and Dongyu Wang [2],***

[1] State Grid Hebei Information & Telecommunication Branch, Shijiazhuang 050013, China; xtgs_yindd@he.sgcc.com.cn (D.Y.); xtgs_chengsy@he.sgcc.com.cn (S.C.); xtgs_zhaow@he.sgcc.com.cn (W.Z.)

[2] School of Artificial Intelligence, Beijing University of Posts and Telecommunications, Beijing 100876, China; pbx@bupt.edu.cn (B.P.); yyqiao@bupt.edu.cn (Y.Q.)

* Correspondence: dy_wang@bupt.edu.cn; Tel.: +133-1105-0257

**Abstract:** The KBQA (Knowledge-Based Question Answering) system is an essential part of the smart customer service system. KBQA is a type of QA (Question Answering) system based on KB (Knowledge Base). It aims to automatically answer natural language questions by retrieving structured data stored in the knowledge base. Generally, when a KBQA system receives the user's query, it first needs to recognize topic entities of the query, such as name, location, organization, etc. This process is the NER (Named Entity Recognition). In this paper, we use the Bidirectional Long Short-Term Memory-Conditional Random Field (Bi-LSTM-CRF) model and introduce the SoftLexicon method for a Chinese NER task. At the same time, according to the analysis of the characteristics of application scenario, we propose a fuzzy matching module based on the combination of multiple methods. This module can efficiently modify the error recognition results, which can further improve the performance of entity recognition. We combine the NER model and the fuzzy matching module into an NER system. To explore the availability of the system in some specific fields, such as a power grid field, we utilize the power grid-related original data collected by the Hebei Electric Power Company to improve our system according to the characteristics of data in the power grid field. We innovatively make the dataset and high-frequency word lexicon in the power grid field, which makes our proposed NER system perform better in recognizing entities in the field of power grid. We used the cross-validation method for validation. The experimental results show that the F1-score of the improved NER model on the power grid dataset reaches 92.43%. After processing the recognition results by using the fuzzy matching module, about 99% of the entities in the test set can be correctly recognized. It proves that the proposed NER system can achieve excellent performance in the application scenario of a power grid. The results of this work will also fill the gap in the research of intelligent customer-service-related technologies in the power grid field in China.

**Keywords:** named entity recognition; knowledge—based question answering; power grid; smart customer service system; SoftLexicon; BERT model; word embedding



## 1. Introduction

Alan Mathison Turing put forward the concept of the "Turing test" in his famous paper "Computing Machinery and Intelligence" published in 1950 to judge whether a machine interacting with human has human intelligence [1]. At present, although the system that completely passes the "Turing test" is still far away from us, there are still some high- performance human–computer interaction products appearing in the market, such as Siri [2], ALIME [3], NetEase Qiyu [4], etc. The KBQA system is the core component of these products. Generally, these voice assistants or intelligent customer services should first use the NER module to extract the topic entity, such as a proper noun, a person's name or the name of an organization, from the text information or voice information input by the user. Then, the retrieval will be built around this extracted entity to generate the answer to users'

query. Therefore, the accuracy of the NER module largely determines the performance of the whole system [5].

Traditional NER methods rely on artificial features and are difficult for mining deep semantic information, which leads to poor NER model performance when an OOV (Out of Vocabulary) problem occurs [6]. In recent years, with the continuous development of deep learning technology, DL-based (Deep Learning-based) NER methods have gradually become mainstream. Compared to a traditional one, a DL-based NER can learn more semantic features through complex non-linear transformation. In 2008, the deep NN (Neural Network) architecture proposed by Collobert and Weston introduced the neural network architecture into the NER task for the first time [7]. The large-scale application of this technology has greatly improved the accuracy and efficiency of NER. In 2013, Mikolov et al. [8] proposed the famous CBOW (Continuous Bag of Words) and Skip-grams models, which led to the widespread adoption of such word-embedding methods when dealing with NLP (Natural Language Processing) tasks. Huang et al. [9] proposed a Bi-LSTM-CRF model, which is robust and has less dependence on word embedding. This model produced the SOTA (State Of The Art) accuracy on several NLP tasks. In 2017, the Google team proposed the Transformer model [10], which made extensive use of self-attention mechanisms to learn text representations with spectacular results. This mechanism has also been widely used in NER tasks since then.

Compared to English NER tasks, Chinese NER is more difficult, since Chinese sentences cannot be naturally segmented by space like English. For Chinese NER, a common practice is to use the CWS (Chinese Word Segment) tool for word segmentation before applying word sequence labeling. However, the CWS tool cannot guarantee that all segmentations are completely correct. These potential errors can greatly affect the performance of the NER model. Despite all this, researchers have proposed some effective methods to solve such problems. Zhang (2018) [11] proposed a Lattice-LSTM structure model which makes full use of character information and word information in Chinese text. This model reduces segmentation errors through lexicon matching. In 2020, Qiu's team proposed a FLAT (Flat-Lattice Transformer) [12] model to improve the disadvantages introduced by the use of an RNN (Recurrent Neural Network) in a Lattice-LSTM. At the same year, Peng et al. [13] proposed the SoftLexicon model based on the Lattice-LSTM. The Softlexicon model simplifies the complicated architecture of the Lattice-LSTM, making it easier to deploy to various downstream NLP tasks without affecting the structure of original system frameworks.

In this work, we focus on the implementation of the Chinese NER task in the KBQA scene. We choose the commonly used Bi-LSTM-CRF model to construct the main part of the whole NER model. At the same time, considering the negative effect brought by incorrect word segmentation, we apply the SoftLexicon model to solve it. However, the SoftLexicon model only shows good performance on MSRA [14], Weibo [15], OntoNotes [16] and other public domain datasets. Whether it can be applied to a domain-specific NER task needs to be verified. Therefore, in addition to utilizing the open-source Chinese public domain dataset NLPCC2016 [17], we also utilize the power-grid-related dataset collected by the Hebei Electric Power Company for experiments. The result shows that the performance of the SoftLexicon-based NER model can be improved by expanding the matching lexicon according to the high-frequency words of the application domain.

In a typical KBQA system, the entities recognized by the NER model will be used to establish a connection between the knowledge base and the user input query. If the entities recognized by the NER model cannot be retrieved in the knowledge base, the knowledge-base-based queries cannot be carried out. Therefore, a complete KBQA system should have the function of fuzzy matching so that those incorrectly recognized entities can retrieve their relevant information in the knowledge base. Some researchers choose to realize this function by introducing deep learning methods [18,19]. For example, Francis-Landau et al. [20] proposed a method that uses CNN (Convolutional Neural Networks) to learn the vector representation of text and then uses cosine similarity scores between

candidate entity vectors and text vectors to realize the matching function. In this work, we regard the fuzzy matching function as a continuation of NER. According to the analysis of NER model recognition results and specific application scenarios, we propose a fuzzy matching module based on the combination of multiple methods. This module combines artificial rules with deep learning methods, which can efficiently recall the entity set with high similarity to the entity to be matched from the knowledge base. The user or system can select an entity from the set to modify the recognition result of the NER model, thereby improving the performance of the named entity recognition module.

The contributions of this work can be summarized as follows:

- We verify the performance of the SoftLexicon+Bi-LSTM-CRF model in NER tasks under a KBQA scenario. Moreover, we explore the applicability of this model in the non-public field, such as a power grid field, and improve the SoftLexicon method according to the application domain.
- To improve the performance of NER, we propose an efficient fuzzy matching module that can modify those entities incorrectly recognized by the NER model based on the combination of multiple methods. This module can be easily deployed in a KBQA system and has strong portability and robustness.
- We further build a dataset and lexicon related to a power grid based on the data provided by the Hebei Electric Power Company and use them to construct an NER system suitable for the power grid field.

The experimental results show that the accuracy of the improved SoftLexicon+Bi-LSTM-CRF model on the power grid dataset is 91.24%, the recall rate is 93.65% and the F1-score is 92.43%. After processing the recognition results by using the fuzzy matching module, about 99% of the entities in the test dataset can be correctly recognized. The experiment results prove that the system achieves good performance on the NER task of the power grid field. Moreover, this system, including the power grid dataset and the power grid lexicon produced by us, can be applied to the construction of a KBQA module of power-grid-related smart customer service system.

## 2. Related Works

### 2.1. Power Grid Intelligent Customer Service System

As mentioned in the Introduction, due to the characteristics of Chinese, the development of a Chinese intelligent customer service system started late. At present, in the public domains of finance, e-commerce and education, some excellent Chinese intelligent customer service systems, such as Tencent Qidian [21], Sobot [22] and so on, have emerged. According to statistics, the market size of China's intelligent customer service industry reached CNY 78.8 billion in 2019, an increase of about 10.06% compared with 2018 [23]. More than 70% of companies have applied Chinese intelligent customer service systems to serve more than 100 million customers.

However, the construction of some non-public intelligent customer service systems in China is still in the initial stage. Taking the field of the power grid as an example, according to the report provided by the Hebei Electric Power Company, the customer service hot line of the power grid company is generally busy during peak hours, and the calls dialed by users cannot be connected. The introduction of an intelligent customer service system can effectively solve the problems existing in the current manual customer service hot line. Due to the limited application scenarios, at present, the research on power grid intelligent customer service systems mainly focuses on overview [24,25], system design [26], technical investigation [27,28] and so on. For some key modules in the system, such as an NER module, no specific technical solutions have been proposed, and few experiments have been carried out using data related to the Chinese power grid field because there are no open-source datasets. We convert the original power grid data collected by the Hebei Electric Power Company into the format that can be used for deep learning training and extract a lexicon of high-frequency words related to the power grid field. We use the above datasets to fine-tune the NER model so that it can be applied to NER tasks in the power grid field.

### 2.2. Text Matching

The fuzzy matching module proposed in this paper is constructed on the basis of a text matching algorithm. The purpose of text matching is to determine whether sentence pairs are semantically similar. Traditional text matching algorithms, such as TF-IDF (Term FrequencyInverse Document Frequency) [29], BM25 [30], Edit Distance [31], Simhash [32], etc., are mainly unsupervised. For instance, the TF-IDF algorithm calculates TF-IDF weight by evaluating the importance of a word to a document and judges the similarity between texts by calculating cosine similarity. This algorithm is efficient and interpretable, but the matching result is not ideal when dealing with complex texts because it only considers word frequency and does not highlight the deep semantic information.

In recent years, with the development of deep learning technology, more and more researchers choose to use deep learning models to solve text matching tasks. Deep learning models excel at using context information to mine deep semantic information, which can effectively solve the shortcomings of traditional algorithms [33]. For example, Arora et al. [34] proposed an unsupervised SIF (Smooth Inverse Frequency) method in 2016. This method represents sentence vectors by the weighted average of word vectors generated by Glove [35], Word2Vec [8] and other methods and then modifies these sentence vectors by PCA (Principal Component Analysis) method. These sentence vectors can be used for textual classification, textual matching and other tasks. Moreover, the tBERT model proposed by Peinelt et al. (2019) [36] predicts semantic similarity by combining semantic features extracted by the BERT model with topic features extracted by the Topic model. This model is prominent in domain-specific cases.

In this experiment, consider the following three features:

- The KBQA system needs to achieve efficient query.
- For the data used in this work, the length of entities is relatively short, normally no more than 10 characters.
- The query text input by the user in a KBQA system generally does not contain contextual information.

Based on the above analysis, we choose to use the unsupervised Word2Vec method to implement a fuzzy matching module. We generate the word embeddings of the words by using the Word2Vec algorithm and obtain the sentence embeddings of entities through the accumulative average method. Then, we calculate the cosine similarity between vectors (sentence embeddings) for matching.

In addition, according to the analysis of the NER model recognition results, we find that even those entities that are not correctly recognized by the NER model usually contain some useful information. Especially for the KBQA system, the query sentence input by the user is relatively short, and the topic entity of the sentence is generally unique. This feature makes the recognition results contain richer effective information. For example, a user enters the query "贝拉克·奥巴马的妻子是谁"? (Who is Barack Obama's wife?). The topic entity extracted by the NER model is "贝拉克·奥巴马" (Barack Obama). However, the knowledge base only stores information about the topic entity "贝拉克·侯赛因·奥巴马" (Barack Hussein Obama). Although the two entities are very similar, they cannot be matched due to the one word (侯赛因 "Hussein") difference. For these entities, using the traditional Edit Distance algorithm for similarity comparison can achieve better results. Therefore, in this work, we combine the DL-based text matching algorithm with the traditional text matching algorithm to construct the fuzzy matching module. Moreover, we add additional artificial rules according to the application scenario to improve the efficiency of fuzzy matching.

## 3. Model and Approach

In this experiment, we use an improved Softlexicon+Bi-LSTM-CRF model to complete the NER task, and propose a fuzzy matching module based on the fusion of multiple methods to modify the incorrectly recognized entities. Among these methods, the Bi-LSTM-CRF model is a commonly used deep learning model in Chinese NER tasks. It can obtain

bidirectional long-term dependencies in the sequence and has a strong nonlinear fitting ability which can be used to model for complex tasks. Compared with the traditional HMM (Hidden Markov Model), the MEMM (Maximum Entropy Markov Model) and other machine learning methods, the Bi-LSTM-CRF model has better performance and robustness [37,38]. The SoftLexicon method makes full use of word and word sequence information while avoiding the negative effect of word segmentation errors. This method can be easily implemented by modifying the character representation layer of existing NER models such as the Bi-LSTM-CRF model. In this way, the performance of the Bi-LSTM-CRF model can be effectively improved on some tasks. At the same time, we also make improvements to this method for specific application scenarios.

However, no NER model can achieve 100% recognition accuracy. In the application scenario of KBQA, the recognition results of an NER model always contain some entities that cannot be retrieved in the knowledge base due to various reasons. As a result, the user's query cannot be connected with the knowledge base, and subsequent work such as entity disambiguation and relationship recognition cannot be carried out. For these entities, we propose a fuzzy matching module suitable for the KBQA system. This module can improve the performance of the NER system by modifying these entities into high similarity entities which can be retrieved in the knowledge base. We first generate the candidate entity set by constructing artificial rules to reduce the matching scope. Then, we calculate the similarity through the text matching method based on the Word2Vec algorithm and the Edit Distance algorithm, respectively, and select the two entities with the highest similarity from each calculation result to form the final entity set. The user or system can select a topic entity from the entity set to modify the incorrect recognition result of the NER model. According to experimental results, for the two datasets used in this work, the accuracy rate of entity recognition will be improved to about 99% after processing with the module.

We combine the NER model with the fuzzy matching module into a NER system. The system can read the query sentence and identify the topic entity of the query. The whole architecture of the system is shown in Figure 1.

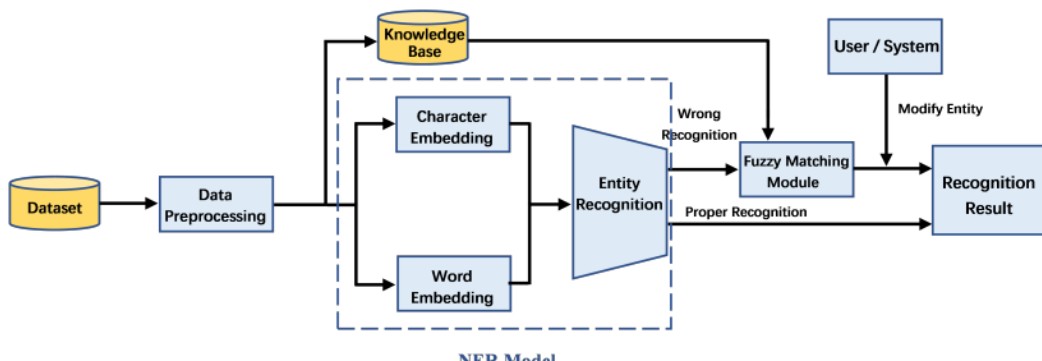

**Figure 1.** The overall architecture of the NER system.

First, we preprocess the obtained data and utilize these data to generate a simple knowledge base. The processed data are input into the NER model for training and prediction. We will try to retrieve the entities recognized by the NER model in the knowledge base (exact matching). For entities that cannot be retrieved from the knowledge base in the recognition result (wrong recognition in Figure 1), the fuzzy matching module will generate an entity set and provide it to the user or the system for selection. The selected entity will be the topic entity of this query.

### 3.1. The NER Model

3.1.1. SoftLexicon Method

SoftLexicon is proposed to solve the long-standing problem of the Chinese NER task. When performing an NER task, words in the context need to be labeled first. An English

sentence can be segmented by space, while a Chinese sentence cannot be segmented directly due to the characteristics of the language. Therefore, researchers tends to use a CWS (Chinese Word Segment) tool for word segmentation first and then use a word-based sequence labeling model to label the segmented sentence [13]. In this process, due to the complexity of Chinese, it is impossible for the CWS tool to precisely segment each sentence. These segmentation errors will affect subsequent model training and entity prediction.

To avoid these problems, some researchers conduct Chinese NER at the character level [39], but this will discard the latent word information contained in the sentence. Therefore, Huang et al. (2018) integrated latent word information into a character-based NER model and proposed a Lattice-LSTM model, which can make full use of the information of word and word sequence without suffering segmentation errors. This model achieved SOTA results on some datasets at the time. However, the speed of model training was slowed down due to the insertion of additional word information into the input sequence. Moreover, the special structure of Lattice-LSTM is difficult to apply to other neural network models [11]. Therefore, Ma et al. (2020) simplified the architecture of the model and proposed the SoftLexicon model. Compared to the Lattice-LSTM, the SoftLexicon method has faster inference speed and is easier to implement.

The SoftLexicon method operates directly on the sequence representation layer of the NER model. For example, if the text is labeled with the "BMES" labeling method, each character $c_i$ in a sentence that matches the lexicon will be classified into four word sets: "B" (Begin), "M" (Mediate), "E" (End) and "S" (Single). For words that cannot match the lexicon, the special symbol "None" will be filled in the word set (Figure 2). The process can be formulated as follows:

$$
\begin{aligned}
B(c_i) &= \left\{ w_{i,k}, \forall w_{i,k} \in L, i < k \le n \right\} \\
M(c_i) &= \left\{ w_{i,k}, \forall w_{i,k} \in L, 1 \le j < i < k \le n \right\} \\
E(c_i) &= \left\{ w_{j,i}, \forall w_{j,i} \in L, 1 \le j < i \right\} \\
S(c_i) &= \left\{ c_i, \exists c_i \in L \right\}
\end{aligned}
\tag{1}
$$

For a sentence $s = \{c_1, c_2, \ldots\ldots, c_n\}$, the $c_i$ in Formulas (1) represents each character in $s$. The $w_{i,k}$ represents the sub-sequence of $s$ (the character set of $c_i$ to $c_k$). $L$ is the lexicon used in the experiment.

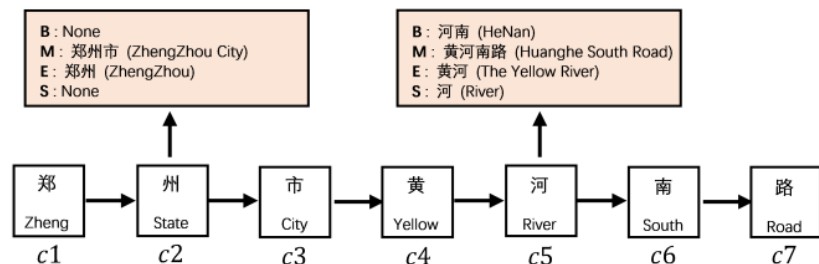

**Figure 2.** An example of generating the "BMES" word set [13]. The input sentence is "郑州市黄河南路" (Zhengzhou Huanghe South Road). The light orange boxes are the "BMES" word sets of the characters "州" (state) and "河" (river).

After obtaining the word set of each character, each word set will be compressed into a fixed-dimension vector by using the pooling method, and the frequency of each word in the dataset will be used as the weight to adjust the result.

$$
v^s(B) = \frac{4}{Z} \sum_{w \in B} z(w) e^w(w)
\tag{2}
$$

Here, $v^s(B)$ is the vector representation of the word set $B$, and $z(w)$ denotes the occurrence frequency of the matched word $w$ in the dataset. The symbol $e^w(w)$ is the word

embedding lookup table [40], which provides the pre-trained word embeddings. $Z$ is the sum of all $z(w)$.

Then, the word set vector $[v^s(B), v^s(M), v^s(E), v^s(S)]$ of each character $c$ and its character embedding $h_c$ obtained by the language model are connected to form the final representation of each character $e_c$ (Formulas (3)). These data can be input into the Bi-LSTM layer for training or prediction.

$$e_c = [h_c; v^s(B), v^s(M), v^s(E), v^s(S)] \tag{3}$$

### 3.1.2. The Bi-LSTM-CRF Model

In this paper, we apply Bi-LSTM to carry out sequence encoding. Bi-LSTM utilizes a bidirectional LSTM network, which can take full advantage of the long-term dependencies from both input directions. The LSTM network is composed of several memory cells (Figure 3).

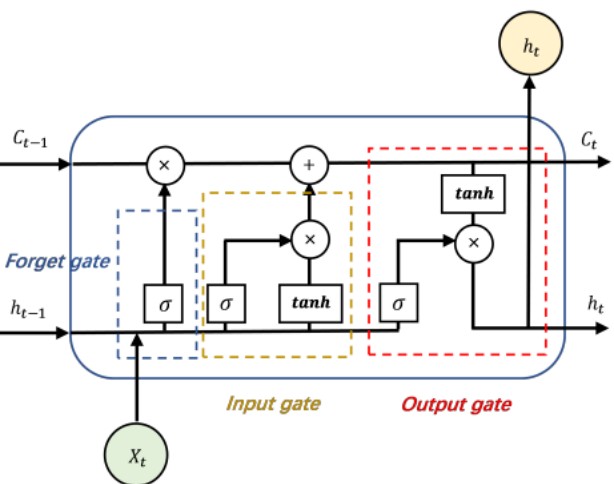

**Figure 3.** The general structure of the LSTM memory cell.

The memory cell is implemented as follows:

$$
\begin{aligned}
i_t &= \sigma(W_i \cdot [h_{t-1}, X_t] + b_i) \\
f_t &= \sigma(W_f \cdot [h_{t-1}, X_t] + b_f) \\
c_t &= f_t * c_{t-1} + i_t * tanh(W_c \cdot [h_{t-1}, X_t] + b_c) \\
o_t &= \sigma(W_o \cdot [h_{t-1}, X_t] + b_o) \\
h_t &= o_t * tanh(c_t)
\end{aligned}
\tag{4}
$$

where $i, f, c$ and $o$ are input gate, forget gate, cell state and output gate; $\sigma$ is the sigmoid function. $W$ and $b$ is the parameter matrix; $h_t$ is the hidden state vector. The output of each LSTM layer can be represented as $H = \{h_t | t = 1, 2, 3 \ldots \ldots, n\}$; $n$ is the length of the input sequence. The hidden state sequences of forward LSTM and backward LSTM are concatenated to form the final sequence representation $H = [\overrightarrow{H_f}, \overleftarrow{H_b}]$ [41].

For the last layer of the NER model, we add a CRF (Conditional Random Field) layer to perform label prediction of the input sequence. CRF combines the characteristics of MEMM and HMM [42]. It is a typical sequence labeling algorithm. In this experiment, the Bi-LSTM layer will process the character representation of the input sequence generated by the SoftLexicon method, and the CRF layer will predict the label according to the output of the Bi-LSTM layer.

### 3.1.3. SoftLexicon+Bi-LSTM-CRF Model

In this experiment, we make some improvements to the SoftLexiocn+BI-LSTM-CRF model according to the specific application scenario. Since the SoftLexicon method directly adjusts the sequence representation layer of the NER model, the structure of the Bi-LSTM-CRF model does not need special changes. The input sentences will be processed by the sequence representation layer using the SoftLexicon method. The generated character representations will be put into the Bi-LSTM-CRF model to obtain the prediction results. The overall structure is shown in Figure 4.

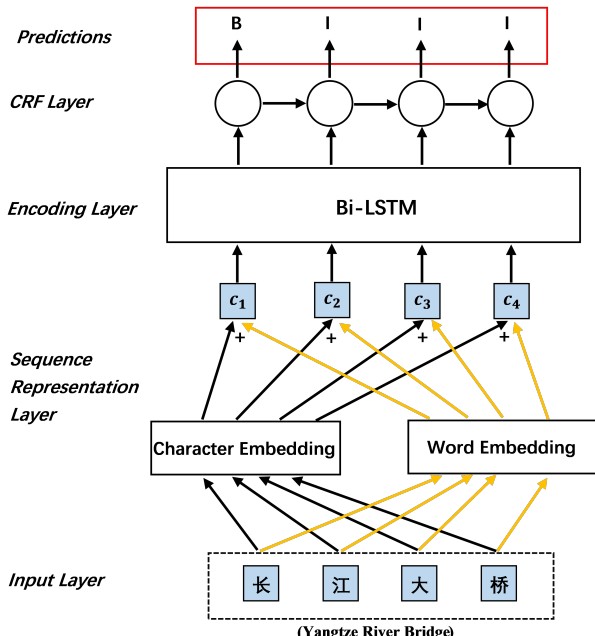

**Figure 4.** The overall architecture of the SoftLexicon+Bi-LSTM-CRF model. (The SoftLexicon method is used to generate word embedding at the sequence representation layer of the model).

We use the commonly used BERT (Bidirectional Encoder Representation from Transformers) model to generate character embeddings [43]. The BERT model used a large amount of corpus for pre-training. In this work, we only need to use the corresponding labeled data (NLPCC2016 and power grid dataset) to fine-tune it. Meanwhile, we utilize the SoftLexicon method to generate word embeddings. In the original paper of the SoftLexicon method, the matching lexicon *L* used by the author is trained from the Chinese Gigaword Fifth Edition corpus [44]. The data sources of this corpus include the newswire from the Xinhua News Agency, articles from Sinorama Magazine, news from the website of the Hong Kong Special Administrative and so on. In other words, the content of the corpus comes from the public domain. The NLPCC2016 used in this experiment is a Chinese open domain question answering dataset, and its corpus is also from the public domain. Therefore, we can continue to use this matching lexicon in the NLPCC2016 dataset-related experiments.

However, another dataset used in this paper, the power grid dataset, is a specific domain dataset. Considering that there are a large number of power-grid-related proper nouns in this dataset, still using the above matching lexicon will cause some proper nouns not to be correctly recognized. Therefore, we clean and screen the data in the power grid database to obtain high-frequency proper nouns related to the power grid domain. These high-frequency words will be used to expand the matching lexicon. At the same time, we use the Word2Vec method to generate word embeddings of these high-frequency words and add them to the word embedding lookup table.

*3.2. Fuzzy Matching Module*

Although above 90% of the topic entities in questions can be recognized correctly by the model proposed in our paper, we find that there are some simple but effective ways to further improve its accuracy. Taking the NLPCC2016 dataset as an example, we divide the causes of topic entity recognition errors into three main cases:

1.  The topic entity cannot be recognized normally due to some rarely used Chinese characters. The BERT model relies on a vocabulary when encoding tokens [45]. For those tokens that cannot be retrieved in the vocabulary (rarely used characters), they will be replaced by a special identifier "[UNK]". The original tokens' information will be discarded, making it impossible to exactly identify the topic entity. Although we can avoid some errors by expanding the vocabulary, it is impossible to add every rarely used Chinese character to the vocabulary in practical application.
2.  The recognized topic entity cannot be linked to the entity in the knowledge base due to typos in user input. For KBQA-related systems, it is a common phenomenon that the user input contains typos. This leads to the fact that even if the entity boundary in the sentence is correctly demarcated, the extracted topic entity cannot exactly link to the knowledge base.
3.  The NER model incorrectly recognizes the entity. No NER model can achieve 100% entity recognition accuracy. Inevitably, there will be some errors in recognition results due to the error of the model.

Aiming at the above three kinds of topic entity recognition errors, we propose a fuzzy matching module to correct them. The construction of the module mainly relies on the following two features:

(1) According to the analysis of the experimental results of the NER model, we find that most of the incorrect entities contain some effective information in either of the above cases. For example, consider the question "你知道中国工服是什么时候出现的吗"? (Do you know when Chinese Kung Fu came into being?). The user wants to ask the question about "Chinese Kung Fu" but enters "Chinese Uniform" when entering the query. Our model can extract the topic entity "中国工服" (chinese uniform) from the sentence, but the right word is "中国功夫" (Chinese kung fu). The recognized entity cannot be retrieved in the knowledge base because of typos, but the word "中国" (Chinese) is an important part of the right word "中国功夫" (Chinese kung fu). We can utilize this effective information to construct a fuzzy matching module.

(2) In the application scenario of the KBQA system, the scope of the user's query is limited. In other words, the user can only query the content that is already stored in the knowledge base. This means when matching, the entity to be matched is obtained from the user's query, while the match objects can only be entities that already exist in the knowledge base. The determination of matching scope enables us to improve the efficiency of fuzzy matching by constructing artificial rules.

Therefore, according to the above two characteristics, we propose a fuzzy matching module based on the combination of multiple methods. With the help of the information in the knowledge base, this module can fix errors in the recognition results of the NER model. In the following part, we will introduce the construction method of the fuzzy matching module.

For a relatively large knowledge base, such as the NLPCC2016 dataset (43,063,796 pieces of data and 5,928,836 entities in total), it is unrealistic to match the recognized entities with each entity stored in the knowledge base. Therefore, an entity candidate set needs to be generated through primary screening first. Here, we will propose a simple but effective method based on artificial rules to generate the candidate set. First, to simplify the knowledge base, we use a CWS tool [46] to segment all entities recorded in the knowledge base. For word segmentation results, the repeated words are combined into a set, and the ID numbers of their corresponding triples in the knowledge base are recorded as the ID set. In addition, considering the efficiency of the matching, we removed words with

a word frequency more than 300 and words with a word frequency less than 3 in the segmented results. After processing the knowledge base using the above method, we obtain a simplified word segmentation dictionary. An example of the process of generating the word segmentation dictionary is shown in Figure 5.

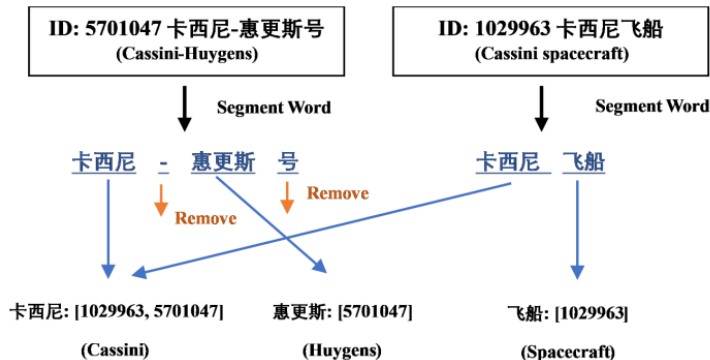

**Figure 5.** An example of the process of generating the word segmentation dictionary. (Words "卡西尼-惠更斯号" (Cassini–Huygens) and "卡西尼飞船" (Cassini Spacecraft) are entities stored in the knowledge base. The [1029963, 5701047] is the ID set of the segemented word "卡西尼" (Cassini), and so on).

Then, we use the CWS tool to segment the entities to be matched. The result of word segmentation will be matched precisely with the words in the word segmentation dictionary. If the match is successful, all entities corresponding to the ID numbers recorded in the word segmentation dictionary will be recalled to construct a candidate entity set. For an entity to be matched $e = \{c_1, c_2, c_3 \ldots c_n\}$, $c$ is the character of the entity $e$.

$$C_{entity} = \sum_{w \in D} \sum_{ID \in IDs} e_{kb} \tag{5}$$

$C_{entity}$ is the generated candidate entity set; $w$ is a word of the entity segmentation results $e = \{w_1, w_2, w_3 \ldots w_n\}$. $D$ is the word segmentation dictionary obtained by processing the knowledge base. *IDs* denotes the *ID set* of the matched word recorded in the word segmentation dictionary; $e_{kb}$ is the entity corresponding to the *ID* in set.

Through the above primary screening method, we narrow the matching scope of the fuzzy matching module. All the entities stored in the knowledge base containing partial information of the entities to be matched are extracted to form the candidate entity set. In the following part, we will determine the final topic entity by comparing the similarity between the entity to be matched and the entity in the candidate set. According to the discussion in the Related Work section, we will use the Word2Vec method and the Edit Distance algorithm to complete the calculation and comparison of similarity.

For the nWord2Vec method, we first use the word embedding averaging method to obtain the sentence embedding of the entity to be matched. For an entity $e = \{w_1, w_2, w_3 \ldots w_n\}$ that has been segmented using the word segmentation tool, we will generate the word embedding of each word $w_i$ according to the word embedding lookup table $e^w$. Then we obtain the sentence embedding $S(e)$ of the entity by averaging the sum of these word embeddings.

$$S(e) = \frac{1}{n} \sum_{i=1}^{n} e^w(w_i) \tag{6}$$

Finally, we calculate the cosine similarity between the sentence embedding of the entity to be matched and the sentence embedding of each entity in the candidate set. In this work, we choose the two entities with the highest similarity in the candidate entity set as the output of the Word2Vec-based matching method.

For the Edit Distance algorithm, we use the FuzzyWuzzy tool to complete the comparison of similarity [47]. The FuzzyWuzzy tool utilizes the Edit Distance (the minimum

number of single-character edits required to change one word into the other) to calculate the differences between sequences. This algorithm has efficient performance in calculating the similarity between short texts. In this part of the work, we similarly choose the two entities with the highest similarity in the candidate entity set as the output of the Edit-Distance-based matching method.

Then we combine the entities obtained from the two methods to form a final entity set. Duplicate entities obtained in both methods will be merged. The entity set will be returned to the user for selection when the entity recognition error occurs. In this way, the fuzzy matching module can modify those incorrectly recognized entities. Alternatively, the system can also select the entity with the highest similarity as the topic entity of this query and perform the following operations such as entity disambiguation and entity linking. An example of a query is in Figure 6.

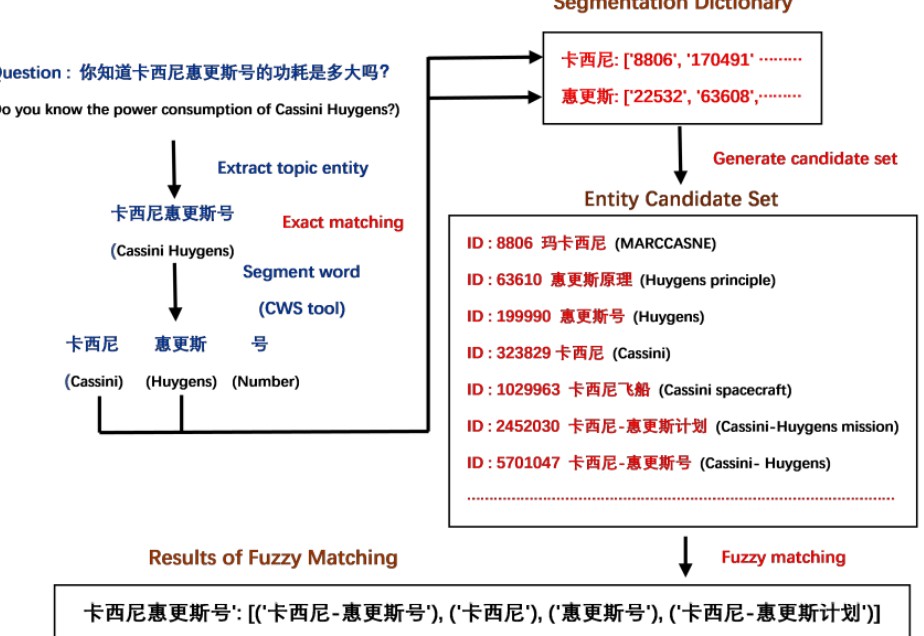

**Figure 6.** The workflow of the fuzzy matching module. The input query is "你知道卡西尼惠更斯号的功耗是多大吗" (Do you know the power consumption of Cassini–Huygens?). The topic entity extracted by the NER model is "卡西尼惠更斯号" (Cassini–Huygens). However, the entity "卡西尼惠更斯号" (Cassini–Huygens) cannot be retrieved in the knowledge base. Therefore, we put it into the fuzzy matching module for further processing. After processing, the correct entity "卡西尼-惠更斯号" (Cassini–Huygens) with the highest similarity to it is included in the candidate results. At the same time, three related entities with high similarity are also included in the candidate results, which improves the robustness of the module.

## 4. Experimental Results and Discussion

### 4.1. Dataset

In this experiment, we use two knowledge bases as experimental data, NLPCC2016 dataset and a power grid dataset, respectively. KB is a structured database that contains a collection of triples in the form of (subject, relation, object) [48]. It is the core of the KBQA system. The system will retrieve the knowledge base according to the query input by the user and return the results of this query.

#### 4.1.1. NLPCC2016

The NLPCC2016 is provided by the NLPCC-ICCPOL (Conference on Natural Language Processing and Chinese Computing) in 2016 [17]. The dataset is made up of two parts. The first part is a large knowledge base that stores a total of 5,928,836 entities and 43,063,796 triples of the entities. The second part is the question–answer pair data.

The NLPCC-ICCPOL divided those into two sets: a training set and a test set. The training set has 14,609 pieces of question–answer pairs, and the test set contains 9870 pieces of question–answer pairs. An example of the dataset is in Table 1.

**Table 1.** Sample data from the NLPCC2016 dataset. The upper part is the sample of triple in the knowledge base. The second part is the sample of the question–answer pair.

| | |
|---|---|
| **Knowledge Base** | 安德烈 ‖ 国籍 ‖ 摩纳哥 <br> **(Andre ‖ Nationality ‖ Monaco)** |
| **Question–Answer Pairs** | **\<question id=4\>** 安德烈是哪个国家的人呢? <br> (What's Andre's nationality) <br> **\<triple id=4\>** 安德烈 ‖ 国籍 ‖ 摩纳哥 <br> (Andre ‖ Nationality ‖ Monaco) <br> **\<answer id=4\>** 摩纳哥 <br> (Monaco) |

The NLPCC2016 training set only provides unlabeled data. Therefore, we need to label the data in the dataset. We remove some invalid data and further divide the data into a training set, a validation set and a test set. Finally, there are 14,480 pieces of data in the training set, 1910 pieces of data in the validation set and 7690 pieces of data in the test set. In this work, we use the "BIO" tagging scheme (the Softlexicon method is also applicable to this tagging method) to label the data: B-begin, I-Inside and O-Outside.

### 4.1.2. Power Grid Dataset

The power grid dataset contains 4630 pieces of data related to the services and systems of power grid companies. Compared to NLPCC2016, the dataset contains some specialist vocabulary related to the power grid field, such as "网讯通" (Wang XunTong), "ERP订单" (ERP order), "多户人口电价" (Electricity price for multiple households) and so on. After screening, we obtained a total of 4523 available data. Since the amount of the data is relatively small, we divide the whole dataset into a training set (3500 pieces of data) and a test set (1023 pieces of data). Then we use the same method for labeling. An example of the original data is in Table 2.

**Table 2.** Sample data from the power grid dataset. The table shows the original power grid data collected by the Hebei Electric Power Company.

| | |
|---|---|
| **Knowledge directory** | 信息通信运维»应用系统»二级部署系统»信息通信一体化调度运行支撑平台（SG-I6000） <br> Information communication operation and maintenance»Application system»Secondary deployment system »Information and Communication Integrated Scheduling Operation Support Platform (SG-I6000) |
| **Knowledge title** | 省公司通用，业务指导，应用系统，I6000系统，登录后部分模块空白（环境支持） <br> Provincial company general, business guidance, application system, I6000 system, some modules blank after login (environment support) |
| **Knowledge content** | 建议使用谷歌浏览器，使用IE部分模块空白 <br> It is recommended to use Google browser, use IE part of the module blank |

The statistics of two datasets are shown in Table 3.

**Table 3.** Statistics of two datasets. Due to the relatively small size of the power grid data, we do not make a validation set for it.

| Datasets | Data Type | Train Set | Validation Set | Test Set |
|---|---|---|---|---|
| NLPCC2016 | Sentence | 14,480 | 1910 | 7690 |
| | Entity | 14,480 | 1910 | 7690 |
| | Tokens | 225,509 | 31,501 | 126,990 |
| Power Grid | Sentence | 3500 | None | 1023 |
| | Entity | 3564 | None | 1050 |
| | Tokens | 65,769 | None | 19,332 |

### 4.2. Experimental Setting

The setting of hyper-parameters in this experiment is partly consistent with the parameters of the SoftLexicon(LSTM) model in Ma et al.'s paper [13]. However, due to the characteristics of the dataset used in the experiment, we adjusted some of the hyper-parameters based on repeated experiments to ensure that the model achieves the best performance. For NLPCC2016, we use the Adam optimizer with the learning rate $1 \times 10^{-5}$ and the decay 0.01. The hidden size of the LSTM is 200, the dropout rate is 0.5, and the epoch number is 3. For the power grid dataset, the learning rate is $3 \times 10^{-5}$, the hidden size of the LSTM is 100, and the epoch number is 2. For the NLPCC2016 dataset, due to its relatively large amount of data, we use a simple cross-validation method. We used the training set and the test set divided by the NLPCC and additionally divided 10% of the data from the test set as the validation set. For the power grid dataset, we used the five-fold cross-validation method. We divided the 4523 pieces of data into five parts and used the training set and the test set alternately for experiments. The average value of five experiments was taken as the final evaluation metric.

### 4.3. Evaluation Metrics

In this experiment, we will evaluate the model with standard precision, recall rate, and F1-score.

$$
\begin{aligned}
P &= \frac{TP}{TP + FP} \\
R &= \frac{TP}{TP + FN} \\
F1 &= \frac{2 * P * R}{P + R}
\end{aligned}
\tag{7}
$$

### 4.4. Experimental Results and Analyses

4.4.1. The Performance of the NER Model

We first show the performance of the NER model. For the NLPCC2016 dataset, the results of the experiment are shown in Tables 4 and 5.

**Table 4.** Comparison between the SoftLexicon+Bi-LSTM-CRF and the Bi-LSTM-CRF on NLPCC2016 dataset. Both models use BERT to generate embedding.

| | Validation Set | | | Test Set | | |
|---|---|---|---|---|---|---|
| | P | R | F1 | P | R | F1 |
| **Bi-LSTM-CRF (BERT)** | 96.54% | 96.65% | 96.60% | 96.80% | 96.89% | 96.84% |
| **SoftLexicon+ Bi-LSTM-CRF (BERT)** | 96.81% | 96.91% | 96.86% | 97.01% | 97.10% | 97.06% |

According to the experimental results, the SoftLexicon method can improve the performance of the Bi-LSTM+CRF model in handling Chinese NER tasks. Since the dataset is dominated by single-entity queries, the entity recognition accuracy of the model is high, so the improvement brought by the SoftLexicon method is not dramatically evident in this work. However, this method does solve some entity recognition errors caused by word segmentation errors. For the test set data, there are a total of 151 entity errors in the recognition results of the Bi-LSTM+CRF model. By using the model combined with the SoftLexicon method, the number of incorrectly recognized entities due to word segmentation errors was reduced to 140.

**Table 5.** Experimental results of nine different models on the NLPCC2016 dataset. The upper part is the model without using BERT, and the lower part is the model using BERT. The result of all models is from our experiments.

| Models | P | R | F1 |
|---|---|---|---|
| **Bi-LSTM-CRF** [1] | | | |
| with random embedding | **91.42%** | **91.29%** | **91.35%** |
| with pre-trained embedding [2] | **92.20%** | **92.12%** | **92.16%** |
| with SoftLexicon | **93.31%** | **93.21%** | **93.26%** |
| **LGN** [49] | **92.22%** | **92.13%** | **92.17%** |
| **Bi-LSTM-CNNs-CRF** [50] | **92.22%** | **92.79%** | **92.46%** |
| **Lattice-LSTM** | **93.07%** | **92.96%** | **93.02%** |
| **BERT** | | | |
| +Bi-LSTM-Softmax | **96.51%** | **96.66%** | **96.58%** |
| +Bi-LSTM-CRF | **96.80%** | **96.89%** | **96.84%** |
| +Bi-LSTM-CRF (SoftLexicon) | **97.01%** | **97.10%** | **97.06%** |

[1] For the Bi-LSTM-CRF model, the words following "with" represent the method to generate embedding. The term "random embedding" means that the model uses randomly generated embedding. The term "pre-trained embedding" means that the model uses the pre-trained word embedding lookup table to obtain embedding. "SoftLexicon" means that the model uses the embedding generated by the SoftLexicon method. [2] The pre-trained word embedding is trained over automatically segmented Chinese Giga-Word using the Word2vec method [11,44]. It can be obtained from the website [51].

In this experiment, we take the basic character-level embedding Bi-LSTM+CRF model as the baseline. For the basic Bi-LSTM-CRF model, the model that uses character embedding generated by the SoftLexicon method performs better than the model that uses randomly generated character embedding or pre-trained character embedding. Lattice-LSTM, LGN and SoftLexicon all utilize lattice structure and lexicon features, and we regard them as a class of models. Among them, SoftLexicon(LSTM) achieved the highest F1 score of 93.26%. In addition, SoftLexicon only modifies the character representation layer of the model, while Lattice-LSTM modifies the LSTM layer of the model, and LGN adds an additional graph structure to the model. Therefore, SoftLexicon has stronger portability and higher efficiency. Among the models without BERT, the SoftLexicon+Bi-LSTM-CRF model has the best performance.

The addition of the pre-trained model BERT significantly improved the performance of the NER model. In this work, even the baseline model Bi-LSTM-CRF achieves an F1 score of 96.84% after incorporating BERT. In addition, comparing BERT+BiLSTM-CRF and BERT+Bi-LSTM-Softmax, we can find that using CRF has better performance in sequence labeling. Compared with Softmax, CRF fully considers the connection between sequence contexts when labeling and adjusts the results through the learned constraints. For the NLPCC2016 dataset, the SoftLexicon+Bi-LSTM-CRF model combined with BERT has the best performance, with an F1 score of 97.06%.

Then, we utilize the power grid dataset to conduct the experiment. In this experiment, we improve the SoftLexicon according to a specific application scenario. Considering that the source of the corpus of the matching lexicon used by the SoftLexicon method comes from the public domain, for the NER task in the power grid field, we utilize some

high-frequency specialist vocabulary obtained from original data collected by the Hebei Electric Power Company, such as "pms2.0", "停电申请单" (blackout application form), "操作票" (operation ticket) and so on to expand the lexicon. The results of the compared experiment are shown in Table 6.

**Table 6.** Experimental results of two models on the power grid dataset. "Expand Lexicon" denotes whether the model uses the expanded lexicon.

| Model | Expand Lexicon | P | R | F1 |
|---|---|---|---|---|
| SoftLexicon+Bi-LSTM-CRF | YES | 91.24% | 93.65% | 92.43% |
| (BERT) | NO | 90.48% | 92.86% | 91.65% |

According to the experiment results, the model using the expanded lexicon achieves better performance on all three evaluation metrics. Some specialized words can be correctly identified. The comparison between the SoftLexicon+Bi-LSTM-CRF model and other models is shown in Table 7.

**Table 7.** Experimental results of eight different models on the power grid dataset. The upper part is the model without using BERT, and the lower part is the model using BERT. The result of all models is from our experiments.

| Models | P | R | F1 |
|---|---|---|---|
| **Bi-LSTM-CRF** | | | |
| with random embedding | **80.45%** | **78.15%** | **79.28%** |
| with pre-trained embedding | **81.34%** | **78.63%** | **79.96%** |
| with SoftLexicon | **86.51%** | **84.45%** | **85.47%** |
| **LGN** [49] | **87.72%** | **87.98%** | **87.85%** |
| **Bi-LSTM-CNNs-CRF** [50] | **81.30%** | **82.16%** | **81.73%** |
| **Lattice-LSTM** | **86.20%** | **84.64%** | **85.41%** |
| **BERT+SoftLexicon** | | | |
| +Bi-LSTM-CRF | **90.48%** | **92.86%** | **91.65%** |
| +Bi-LSTM-CRF (Expand Lexicon) | **91.24%** | **93.65%** | **92.43%** |

The experimental results show that in some specific fields, such as the power grid, the model combining lattice structure and lexicon features (SoftLexicon, Lattice—LSTM and LGN) has significantly better performance than other models. The application of the BERT model has greatly improved the performance of this kind of model. Among these models, the SoftLexicon+Bi-LSTM-CRF (Expand Lexicon) model achieves the best performance. Therefore, it proves that expanding the matching lexicon used by the SoftLexicon method according to the application domain can further improve the performance of the model. In addition, the SoftLexicon+Bi-LSTM-CRF model can achieve excellent performance on NER tasks related to the power grid domain. This model can be applied to the construction of grid-related intelligent customer service systems.

### 4.4.2. The Performance of the Fuzzy Matching Module

Next, we input the entities incorrectly recognized by the NER model into the fuzzy matching module for an experiment. The experimental results are shown in Table 8. For the NLPCC2016 database, out of 7690 pieces of test data, a total of 225 entities are not correctly recognized. After inputting these entities into the module for fuzzy matching, 135 of them can be modified to the correct entities which can be successfully retrieved in the knowledge base. After processing by this module, the accuracy of the named entity recognition will be further improved to about 99%. Subsequently, we also use the power grid dataset to perform the similar experiments. For the power grid dataset, out of 1023 pieces of test data, a total of 77 entities are not correctly recognized. After inputting them into the

fuzzy matching module, 68 of them can be modified to the correct entities which can be retrieved in the knowledge base. The accuracy of the named entity recognition will be further improved to about 99.12%. Therefore, the experiment results show this module will improve the performance of Chinese NER in the KBQA application scenario. Relying on the information stored in the knowledge base, this module can achieve efficient fuzzy matching. Moreover, the fuzzy matching module has strong portability and is easy to deploy. It can be easily deployed into existing NER models to form an NER system, which can help construct the KBQA system with better performance.

**Table 8.** Experimental results of the fuzzy matching module in the NLPCC2016 dataset and the power grid dataset.

| Dataset | Incorrect Results | Successfully Modified [1] | Response Time [2] | ACC [3] |
|---|---|---|---|---|
| NLPCC2016 | 225 | 135 | 0.78s | 99.0% |
| Power Grid | 77 | 68 | 0.23s | 99.1% |

[1] "Successfully Modified" means that the correct entity of the input query is included in the candidate entity set generated by the fuzzy matching module. [2] "Response Time" means the average response time for one query. [3] "Precision" means the accuracy of the NER system (composed of the NER model and the fuzzy matching module).

## 5. Conclusions

In this work, we focused on the research of the Chinese NER task in the KBQA system. We analyzed the characteristics of input text of the KBQA system, and proposedthe an NER system suitable for KBQA scene. For the NER model, considering the problem of word segmentation in a Chinese NER task, we introduce a SoftLexicon method based on the original Bi-LSTM-CRF model. This method combines the semantic information of characters and words in the textand adopts a lexicon matching method to avoid the negative impact of word segmentation errors on the model. Different from a general NER task, for the KBQA system, we hoped to find a similar entity as the topic entity of the query when the NER model cannot recognize the correct entity to carry out subsequent tasks such as entity disambiguation and relationship recognition. Therefore, we proposed a fuzzy matching module. This module combines a traditional text matching method with a DL-based text matching method to perform fuzzy matching, which improves the robustness of the module. In addition, we used the characteristics of the data in the KBQA system to construct artificial rules to improve the efficiency of fuzzy matching. After processing by this module, for both the NLPCC2016 dataset and the power grid dataset, the accuracy of the NER reached more than 99%.

In addition, using the grid-related data collected by the Hebei Electric Power Company, we carried out effective research on the construction of power grid intelligent customer service. We preprocessed the grid-related original data and tagged the data for the training of a deep learning model. Then, we extracted a lexicon of high-frequency words in the power grid domain from the dataset and used this lexicon to expand the matching lexicon used by the SoftLexicon method. The experimental results show that the improved SoftLexiocn-Bi-LSTM-CRF model has significantly better performance when dealing with NER tasks in the power grid domain.

At present, there are few effective research studies on a power grid intelligent customer service system in China. The NER system and the grid dataset we produced will fill the gap in the research of intelligent customer service system technology in the field of power grids. Our proposed system still has some shortcomings and limitations. For example, the system needs to be implemented with a high-frequency word matching lexicon, so it has a relatively high requirement on the quality of data in the application field. In addition, the huge amount of parameters of the BERT model may affect the efficiency of the system when deployed to the power grid intelligent customer service system. In the future, in view

of the above problems, we will continue to improve the system. Firstly, we will try to use the transfer learning method to reduce the dependence of the model on domain data [52]. Secondly, some research has proposed some simplified BERT models, such as TinyBERT [53], MobileBERT [54] and so on, but they are difficult to use to replace the classical BERT model due to their insufficient generalization ability. In the future, we will explore the applicability of these simplified BERT models in our system. Finally, considering that the general intelligent customer service system not only needs to process text information, but also needs to process multimodal information such as images and sounds, our future research will focus on the technology of multimodal machine learning [55].

**Author Contributions:** Conceptualization, D.Y. and B.P.; methodology, D.Y. and B.P.; software, B.P.; validation, D.Y., B.P. and S.C.; formal analysis, B.P.; investigation, Y.Q. and D.W.; resources, S.C.; data curation, S.C.; writing original draft preparation, B.P.; writing review and editing, D.Y.; visualization, D.Y.; supervision, W.Z.; project administration, Y.Q., D.W. and W.Z.; funding acquisition, W.Z. All authors have read and agreed to the published version of the manuscript.

**Funding:** This research was funded by the science and technology project of State Grid Corporation of China "Research on semantic retrieval and analysis technology of intelligent customer service based on deep learning" (SGHEXT00DDJS2100105).

**Institutional Review Board Statement:** Not applicable.

**Informed Consent Statement:** Not applicable.

**Data Availability Statement:** Data are available upon request.

**Acknowledgments:** We thank the State Grid Hebei Information & Telecommunication Branch for providing the research data and funding support for this paper.

**Conflicts of Interest:** The authors declare no conflict of interest.

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
