# Peer review of "Chinese Named Entity Recognition Based on Knowledge Based Question Answering System"

_applsci, doi:10.3390/app12115373_

Round 1

Reviewer 1 Report

  1. "ralated” – typo.
  2. “(Barack Hussein Obama). Although the two entities are very similar, they cannot be matched due to one character difference.”. Why character? It is a word in English! A character is only in Chinese. You have to say this.
  3. “The whole architecture of the system is shown as follows:”. You should write like “The whole architecture of the system is shown in Figure 1.”
  4. Which boxes are yellow in Fig. 2?
  5. No reference to Fig. 2.
  6. No reference to Fig. 5.
  7. “sampel” – typo.
  8. The term “character-based NER model” is only in the review part of the manuscript and in the conclusion section. Therefore, the presence of this tern in the conclusion section raises many doubts.
  9. “Knowledge Based”. Why do not you write this term as “Knowledge-Based”? In many other cases, we find the terms: “character-based, ‘word-based” and others “-based”. No consistency.
  10. Define the limitations of the proposed method.
  11. You are advised to remove the first two statements from the abstract. A long history is not needed.

Reviewer 2 Report

The manuscript is written well. However, some small revisions need to address before publication: 
(1) The typo for the learning rate value at line 439 
(2) Explain why the number of epochs is so tiny by 2 
(3) Lack of comparison performance of the proposed method with another method in related works

Author Response

Thank you for your kind suggestions and comments.  For Point 1 and Point 3, we will make corresponding modifications in the resubmitted manuscript. 

For Point 2, we research lots of papers like [1], [2] and [3]. Although the paper used different datasets and models, but the setting of epoch number is all small. Different from some traditional models like Bi-LSTM, CNN and son on, the BERT model can extract rich semantic information due to its huge network architecture. In this work, when implementing the fine- tuning training, the loss converges faster, so we set a relatively small epoch number.   At the same time, we have also used other loss functions and epoch numbers for experiments. The results show that the highest F1-score can be obtained when epoch is 2.  When epoch is 3, 4 and higher, loss begins to fluctuate and the F1-score also gradually decreases.  In addition, the small epoch may also be related to the dataset we used. Since the dataset is collected by power grid company in their daily business, it does not contain many rare words and typos. At the same time, the size of dataset is small (about 4500 pieces of data) and each query in the dataset is mainly Chinese short text, and the text length is usually no more than 20 characters. Therefore, the model can converge speedily on the power-grid dataset and achieve good results.

[1] Jia C, Shi Y, Yang Q, et al. Entity enhanced BERT pre-training for Chinese NER[C]//Proceedings of the 2020 Conference on Empirical Methods in Natural Language Processing (EMNLP). 2020: 6384-6396.

[2] Cai Q. Research on Chinese naming recognition model based on BERT embedding[C]//2019 IEEE 10th International Conference on Software Engineering and Service Science (ICSESS). IEEE, 2019: 1-4.

[3] Ma R, Peng M, Zhang Q, et al. Simplify the usage of lexicon in Chinese NER[J]. arXiv preprint arXiv:1908.05969, 2019.

Reviewer 3 Report

The article contains information technical and innovative. The problem addressed is current and has relevance, which makes it significant. The paper is well organized. The experimental methodology is described comprehensively. Interpretations and conclusions are justified by the results. This reviewer has identified the following main issues:

  • The abstract can be rewritten to be more meaningful. The authors should add more details about their final results in the abstract. The abstract should clarify what is exactly proposed (the technical contribution) and how the proposed approach is validated.
  • The paper does not clearly explain its advantages with respect to the literature: it is not clear what is the novelty and contributions of the proposed work: does it propose a new method? Or does the novelty only consists in the application?
  • The paper does not provide significant experimental details needed to correctly assess its contribution: What is the validation procedure used?
  • Quality of Figures is so important too. Please provide some high-resolution figures. 
  • Discuss the future plans with respect to the research state of progress and its limitations.

Reviewer 4 Report

This work proposes a system to automatically answer natural language questions based on NER model and fuzzy matching module. To improve NER results, the system retrieves structured data stored in the knowledge base. In experiments, authors prove the performance compared various baselines such as LGN, Bi-LSTM-CNNs-CRF and Lattice-LSTM. However, authors do not describe some points as following questions.

Strong points

S1. In this works, they compare proposed method with various baselines and existing works.

S2. Chinese language has special character that it is difficult to categorize named entities unlikely English language. Therefore, they acquire the information from knowledge base to improve the NER result.

S3. They demonstrate the performance using real word dataset.

Weak points

W1. Proposed method is interesting, but some points aren’t presented enough. In Fig 1., NER model produces a recognized result. The result is classified “wrong recognition” or “proper recognition”. However, they do not describe how to classify them. Since the system retrieve from knowledge base to utilize fuzzy matching module, the result whether it is wrong or proper is so important. However, there is no description about the mechanism. Does human manually classify it? Or some undescribed module automatically does that?

W2. When exploiting BERT as word embedding, some details on how they pre-train the module are missing in the manuscript. IMHO, NLPCC2016 and Power Grid seem not enough to pre-train BERT.

W3. Some experimental results need to be explained in more detail. For example, In Tab. 5., they replace word embedding module BERT with random embedding or pre-trained embedding. Which pre-trained embedding was used?

Minor comments

Some apostrophes are not properly spaced.

Round 2

Reviewer 3 Report

The authors made all the requested changes. I am in favour of the publication.